# Extended spectrum beta lactamase producing bacteria among outpatients with ear infection at FelegeHiwot Referral Hospital, North West Ethiopia

Kindye Endaylalu[1], Bayeh Abera[2], Wondemagegn Mulu[2]*

1 Department of Microbiology Laboratory, Amhara Public Health Institute, Bahir Dar, Ethiopia,
2 Departmentsof Medical Laboratory Science, College of Medicine and Health Sciences, Bahir Dar University, Ethiopia

* wondem_32@yahoo.com

**Data Availability Statement:** All relevant data are within the manuscript and its Supporting Information files.

**Funding:** This work was supported by Bahir Dar University, College of Medicine and Health

## Abstract

### Background

Antibiotic resistant bacteria particularly extended-spectrum beta lactamase (ESBL) producing are of major concern for management of outpatients. They can spread rapidly and are associated with poor patient outcome. However, there is scarcity of information on ear infection with ESBL producing bacteria in Ethiopia. Therefore, this study investigates the prevalence of ear infection with ESBL producing bacteria among outpatients attending Felegehiwot Referral Hospital, Northwest Ethiopia.

### Methods

A hospital based cross-sectional study was conducted from May, 2018 to January, 2019. Demographic and clinical data were collected with face to face interview and were complemented with patient card review. Ear discharge specimens were collected from study participants using swab technique. All ear swab specimens were cultured using standard microbiological techniques. The ESBL producing bacteria were detected by double disc synergy test and interpreted based on Clinical and laboratory Standards Institute Guidelines. Chi-square and fisher's exact tests were calculated to check association between variables.

### Results

A total of 236 patients (male = 138 and female = 98) with ear infection took part in the study. The median age of the participants was 20years. Overall, 10 (4.23%, 95%CI; 2.3–7.6%) of patients had ear infection with ESBL producing bacteria. Other chronic illnesses (p = 0.003), history of hospital visit and treatment (p = 0.006) and history of antibiotic use without physician's prescription (p<0.001) had significant association with prevalence of ESBL producing bacteria in ear infection. The proportion of ear infection with ESBL producing *P.mirabilis*, *P. aeruginosa* and *K.pneumoniae* were 4 (1.7%), 3 (1.3%) and 2 (0.8%), respectively. All ESBL producing isolates were MDR (100%). Overall, 58 (43%) species were MDR. *P.aeruginosa*

Sciences. The funder had no role in study design, data collection and analysis, decision to publish, or preparation of the manuscript.

**Competing interests:** The authors have declared that no competing interests exist.

was the leading MDR isolate 29 (53.7%).For all bacterial isolates of ear infection, ampicillin (93.3%) and amoxicillin-clavulanic acid (58.5%) revealed high level of resistance whereas low resistance rates were observed for ciprofloxacin (5.2%), third generation cephalosporin (11.9–20%) and aztreonam (16.3%).

## Conclusions

Ear infection due to ESBL producing bacteria coupled with high levels of MDR is becoming a growing concern for outpatients in the study area. Regular detection of these bacteria and wise use of antibiotics are needed to stop the spread of this form of resistance.

## Introduction

Ear infection is a common clinical problem throughout the world and a major cause of hearing loss in developing countries [1]. Otitis externa, otitis media and inner ear infections are the major types of ear infections [2]. Otitis externa is an inflammation of the external auditory canal [2] and otitis media is an inflammation of the middle ear and mastoid process. Otitis media frequently occurs in children [3, 4] and it includes acute otitis media (AOM), otitis media with effusion (OME) and chronic suppurative otitis media (CSOM). Ear infections are closely related but differ in presentation, associated complications and treatment [1, 5]. Globally, 30 per 10,000 people had hearing loss and 21, 000 people die every year due to complication of otitis media [5].

Bacterial cause of acute otitis media (AOM) is characterized by the presence of fluid in the middle ear together with acute ear pain, fever and general illness. It is one of the most frequent causes of antibiotic use in children [1]. Acute mastoiditis, meningitis and brain abscesses are common and potentially serious suppurative complications of AOM [5].

Chronic suppurative otitis media (CSOM) is a chronic inflammation of the middle ear and the mastoid cavity with perforated tympanic membrane and ear discharge persisting for at least two weeks [6, 7]. It is characterized by the presence of an effusion and intact tympanic membrane in the absence of signs and symptoms of acute inflammation. CSOM can lead to hearing impairment, brain abscesses, or meningitis in childhood and late in life [8]. It also results in severe disability and death in developing countries.

*Pseudomonas aeruginosa* (*P.aeruginosa*),*Klebsiella pneumoniae* (*K.pneumoniae*), *Proteus* species,*Haemophilusinfluenzae* (*H.influenzae*) and *Moraxella* are the most common aerobic microbial speceis isolatedfrompatients with otitis media and otitis externa [9, 10]. In Ethiopia, *Proteus* species, *P.aeruginosa*, *K.pneumoniae* and *Escherichia coli* are commonly reported Gram negative bacteria isolatedfrom ear infection [3, 10, 11].

Beta-lactam drugs, fluoroquinolones and aminoglycosides are most frequently prescribed antibiotics to treat bacterial infections [12]. However, the widespread use of these antibiotics has caused the emergence and spread of resistant bacteria [13, 14]. Currently, increasing trends of antibiotic resistance rates to ear infection are reported worldwide in gram negative bacteria. One of the mechanisms of resistance is synthesis of extended spectrum beta-lactamases (ESBL). They are bacterial enzymes capable of hydrolyzing penicillin, first, second and third-generation cephalosporins, and aztreonam, but not the cephamycins or carbapenems [14]. ESBL producing Gram negative bacilli (GNB) are a concern in healthcare setting and in the community [15, 16].

Majority of ESBL producing bacteria in clinical samples are resistant to various antibiotics, leaving only limited drugs as a choice of therapy. They frequently carry genes encoding resistance to aminoglycosides, quinolones and cotrimoxazole. Thus, ESBL producing organisms often possess multidrug resistance phenotypes [14]. In India 18.3–74.2% rate of ear infection with ESBL producing organisms have been reported [16].Similarly, in Nigeria and Tanzania, 8.3% and 16.3% ESBL producing GNB were isolated from ear swab samples, respectively [17, 18].In Ethiopia there are no data on ear infection with ESBL producing bacteria. However, 38.4–57.6% rate of ESBL producing isolates were documented from other clinical and environmental samples [13, 19, 20].Repeated use of antibiotics, chronic disease, prolonged hospitalization, poor hygiene, inadequate health care, recurrent upper respiratory tract infections and overcrowding are frequently documented factors associated with ear infection [21–25].

Ear infection with ESBL producing bacteria is becoming a worldwide serious problem for treatment. Infections caused by ESBL producing bacteria are associated with poor treatment outcome. It is the major cause of treatment failure, increased cost of health care, prolonged hospitalization, increased complications, disability and mortality [18, 26]. However, studies done on the etiologies of ear infections in the world, did not adequately address the burden of ESBL producing bacteria in ear infection. Moreover, the causative agents of ear infection vary in different geographic regions of the world including Ethiopia [27].

Although ESBL production is a significant and devastating clinical problem, most laboratories in Ethiopia do not perform ESBL detection tests for diagnosis or infection control purpose with an eventual empirical treatment. At the same time baseline information on the magnitude of ear infection with ESBL producing bacteria is limited. Studies on the prevalence of ear infection with ESBL producing bacteria are indispensable in order to overcome the spreading of antimicrobial resistance. Therefore, this study was intended to finds out the prevalence of ear infection with ESBL producing bacteria among outpatients with ear infection attending at FHRH Ear, Nose and Throat (ENT) clinic, Northwest Ethiopia.

## Materials and methods

### Study design, period and setting

A hospital based cross-sectional study was conducted from May, 2018 to January, 2019 at ENT clinic of FHRH, Bahir Dar, Northwest Ethiopia. FHRH provides health care services and is a center for medical education and research. It provides healthcare services to at least 7 million populations living in and around Bahir Dar city. FHRH ENT clinic diagnoses and treats a wide variety of conditions of ear, nose, mouth, head, and neck and throat infections with surgical and non-surgical options. Besides, Amhara Public Health Institute (APHI) which is located at Bahir Dar city is a regional reference laboratory where samples were processed. This laboratory is accredited by the Ethiopian National Accreditation Office (ENAO) in 2019.

### Sample size and sampling

The sample size was determined using a single population proportion formula considering estimated proportion of ear infection with ESBL producing bacteria (0.05),95% confidence level and marginal error of5%. Accordingly, the calculated sample size was 384. However, the average daily flow rate of patients with ear infection at FHRH was 3 and a total of 486 patients were expected to visit the hospital during the study period which is less than 10,000. Thus, finite population correction was assumed for proportions and a final sample size of 215 was obtained. However, a total of 236 patients with ear infection were included in the study. All patients who volunteered and fulfilled the inclusion criteria were included conveniently. Thus, consecutive specimens were collected from 236 outpatients with discharging ear at ENT clinic

of FHRH. All ear infected patients with discharging ears at FHRH during the study period were the study populations.

## Variables

Ear infection with ESBL producing bacteria was the dependent variable while demographic variables (age, sex, residence) and clinical variables (type of otitis media, hearing status, color of ear discharge, having chronic disease, presence of upper respiratory tract infection, previous hospital visit and treatment and antibiotic use without physician's prescription)were the independent variables.

## Inclusion and exclusion criteria

Patients of all age groups with clinical evidences of either otitis media or otitis externa that provided pus swab from discharging ears were included in the study. However, patients on antibiotic treatment within the previous two weeks of sample collection and those without discharge from their ear were excluded from the study.

## Data collection

Data on demographic variables and other variables of ESBL producing bacterial ear infection were collected from each study participant by face-to-face interview, medical history and patient card review using structured questionnaire.

## Ear specimen collection and transport

Two ear swab specimens from patients with discharging ear were collected aseptically following swab techniques using sterile cotton-wool by the study team in consultation with the respective physician following bacteriological standard procedures. The swab specimens were immediately placed onto plain test tubes and transported to APHI Microbiology Laboratory within30 minutes of collection [26].

## Bacterial isolation and identification

Standard microbiological techniques were used for isolation and identification of bacteria. [27]. Each ear swab samples were inoculated on Blood agar, Chocolate agar and MacConkey agar (Oxoid, UK) at a time. Blood and chocolate agar plates were incubated in a candle jar which can generate about 5% $CO_2$. All of the inoculated media were incubated at 37°C for 24h. All positive cultures were identified with grown bacterial colony characteristics on the respective media. Each bacterial colony was further characterized by Gram stain and biochemical profiles using standard methods [28]. Members of the family Enterobacteriaceae and other Gram-negative rods were identified by biochemical tests including indole production, lactose fermentation, citrate utilization, gas production, motility, urease test and oxidase tests [27].

## Antibacterial susceptibility testing

Antibacterial susceptibility testing was performed by using Kirby–Bauer disk diffusion method on Mueller Hinton Agar (Oxoid, UK) [27]. All isolates were tested against the following classes of antibiotic disks: Penicillin (ampicillin (10μg)), β-lactam/β-lactamase inhibitor combination (amoxicillin-clavulanic acid (20/10μg)), Folate pathway inhibitor (trimethoprim-sulphamethoxazole (1.25/23.75μg), third generation cephalosporin (ceftriaxone (30μg), ceftazidime (30μg), and cefotaxime (30μg),aminoglycosides (gentamycin (10μg)),fluoroquinolones (ciprofloxacin (5μg)), and monobactam (aztreonam (30 μg) (AbtekBiologicals, UK) [27].Resistance

data were interpreted using the standard zone sizes of the Clinical and Laboratory Standards Institute (CLSI) [27].

## Detection of ESBL producing bacteria

ESBL-producing isolates were detected through phenotypic screening and confirmation steps using the Kirby Bauer disk diffusion test [27]. In the screening step, a 0.5 McFarland matched bacterial suspension was properly inoculated on Mueller Hinton Agar (Oxoid, UK). Cefotaxime(30 μg), ceftriaxone(30 μg), ceftazidime(30 μg) and aztreonam(30 μg) disks were placed at least 30mm apart on the inoculated MHA plate and incubated overnight at 37˚C. Isolates that showed the inhibition zone size ≤ 22 mm for ceftazidime and/or cefotaxime ≤ 27 mm and /or ceftriaxone ≤ 25 mm and/or Aztreonam ≤ 27 mm were considered as ESBL positive [27]. ESBL positive bacterial isolates were then subjected to confirmatory test.

## Confirmation for ESBL producing isolates

All strains which were positive for the ESBL screening test were selected and checked for ESBLs production by using Double Disk Synergy Test (DDST) [28].A suspension of the organisms to be tested was made with normal saline and spread onto MHA plate. Ceftazidime (30 μg), cefotaxime (30 μg), ceftriaxone (30 μg), and aztreonam (30μg) disks were placed around amoxicillin-clavulanic acid (20/10μg) 15mm apart. After overnight incubation at 37˚C, isolates that showed any increase in the zone of inhibition towards the disk of amoxicillin-clavulanic acid was considered as positive for the ESBL production [27].

## Quality control

Questionnaire was prepared in English version and translated in to local language and back to English to check its consistency by a professional translator. A standard bacteriological procedure was followed to keep the quality of all laboratory tests. Sterility of culture media were checked by incubating 5% of the batch at 37˚C overnight and evaluated for possible contamination. American Type Culture Collection (ATCC) standard reference strains (*E.coli*ATCC-25922 and *K.pneumoniae*ATCC-700603) were used as quality controls for ESBL detection [27].

## Data analysis

Data were checked, entered and analyzed using Statistical Package for Social Science 23 (IBMCorpReleased2011.IBMSPSSstatistics. Armonk, NY: IBMCorp). Descriptive statistics was calculated to describe relevant variables. Chi-square test and Fisher's exact test were obtained used to determine association between dependent and independent variables. P-value of <0.05 was considered statistical significant.

## Ethical considerations

Ethical approval was obtained from Bahir Dar University, College of Medicine and Health Sciences, Institutional Review Board (IRB). Written informed consent was obtained from study participants, guardians or caretakers of children after explaining the purpose and objective of the study. Information obtained from participants kept confidential. The laboratory results from the study participants were communicated to their physicians for proper management of patients.

# Results

## Demographic characteristics

A total of 236 patients with ear infection tookpart in the study. Among them, 138(58.5%) were males and 138 (58.5%) had clinical evidences of chronic suppurative otitis media. The ages of study participants ranged from 1 to 64 years with a median age of 20 years. Majority (64%) of the study participants were from rural settings. Among the totalpatients,98 (41.5%) and 108 (45.8%) had infections on the right and left ear, respectively and 128 (54.2%) had decreased hearing status. Majority of cases had white 140 (59.3%) and yellow 57 (24.2%) color ear discharges. History of previous visit to health care facilities linked to ear infection was found among 88 (37.3%) patients and these patients had been treated with antibiotics (Table 1).

## Prevalence of ear infection with ESBL producing bacteria

Overall, the prevalence of ear infection with ESBL producing bacteria was 10(4.23%) and all of them were from rural dwellers. The highest proportion of ear infection4 (14.3%) was recorded among patients from 31–40 years of age ($P$ = 0.032).The proportion of ear infection with ESBL producing bacteria was 7 (5.1%) in males and 3 (3.1%) in females. ESBL production was found in 7 patients with CSOM (5.1%) and 3 patients with AOM (3.2%). The proportion of ESBL production was the highest in patients with other chronic disease 6 (46.2%) ($P$ = 0.003). Ear infection with ESBL producing bacteria was higher in those patients who took non-prescribed antibiotics than those who did not ($P<0.001$). ESBL producing bacterial isolates were higher in those patients who had history of hospital visit for treatment than those who did not ($P<0.006$) (Table 1).

The proportion of ear infection with ESBL producing *P.mirabilis* and *P.aeruginosa* was4 (1.7%) and 3 (1.3%), respectively. On the other hand, ESBL producing *K.pneumoniae* and *E. coli* accounted for2 (0.8%) and 1 (0.4%) of ear infections, respectively(Fig 1).

## The isolation rate and their profile of ESBL production

Overall,135 (57.2%) Gram negative bacterial species were isolated. Among them, *P.aeruginosa* 54(40%) was the most frequent isolate followed by *P.mirabilis* 43(31.9%) and *K.pneumoniae* 16 (11.9%). Mixed Gram negative bacterial infections found in 8 patients. Among them, 5 were *P. aeruginosa+ P.mirabilis* (Table 2). From all isolates, 10 (7.4%) were ESBL producing. The proportion of ESBL producing *K.pneumoniae*, *E.coli*, *P.mirabilis* and *P.aeruginosa* was 2(14.3%), 1 (12.5%), 4 (10.8%) and 3 (6.4%), respectively (Table 2).

## Antimicrobial resistance profiles of Gram negative bacterial isolates

Overall, Gram negative isolates revealed 5.2–93.3% level of resistance to the commonly used antibiotics. The highest resistance rate was found against ampicillin 126(93.3%), amoxicillin with clavulanic acid 79(58.5%) and sulfamethoxazole-trimethoprim 42(31.1%). However, lower resistance rate were observed to ciprofloxacin 7(5.2%), third generation cephalosporins (11.9–20%) and aztreonam (16.3%) (Table 3).

*Pseudomonas aeruginosa*, *P.mirabilis*, *K. pneumoniae* and *E. coli* were resistant to ampicillin (87.5–96.3%) and amoxicillin-clavulanic acid (44.4–74.1%). On the other hand, *P.aeruginosa* showed low rate of resistance for ciprofloxacin 3(5.6%) and cefotaxime 6 (11.1%). *P.mirabilis* also showed low rate of resistance against ciprofloxacin 3 (6.9%), cefotaxime 4 (9.3%) and ceftazidime 5 (11.6%) (Table 3).

**Table 1. Prevalence of ear infection with ESBL producing bacteria in relation to demographic characteristics and other clinical variables of outpatients at FHRH May 2018 to January 2019.** (N = 236).

| Variables | Ear infection with ESBL producing bacteria | | | |
|---|---|---|---|---|
| | Positive N (%) | Negative N (%) | Total N (%) | P-value |
| Age(years) | | | | |
| ≤ 10 | 0 (0) | 60 (100) | 60 (25.4) | |
| 11–20 | 1(1.6) | 62 (98.4) | 63 (26.7) | |
| 21–30 | 2(4.5) | 42 (95.5) | 44 (18.6) | |
| 31–40 | 4 (14.3) | 24 (85.7) | 28 (11.9) | 0.032 |
| 41–50 | 2 (10.0) | 18 (90.0) | 20 (8.5) | |
| ≥ 51 | 1 (4.8) | 20 (95.2) | 21 (8.9) | |
| Sex | | | | |
| Male | 7 (5.1) | 131(94.9) | 138 (58.5) | 0.529 |
| Female | 3 (3.1) | 95 (96.9) | 98 (41.5) | |
| Residence | | | | |
| Urban | 0 (0) | 85 (100) | 85 (36) | |
| Rural | 10(6.6) | 141(93.4) | 151 (64) | NA |
| Type of Otitis media | | | | |
| AOM | 3 (3.2) | 91 (96.8) | 94 (39.8) | NA |
| CSOM | 7 (5.1) | 131(94.9) | 138 (58.5) | |
| Otitis externa | 0 (0) | 4 (100) | 4 (1.7) | |
| Infected ear | | | | |
| Left | 4 (3.7) | 104 (96.3) | 108 (45.8) | 0.180 |
| Right | 6 (6.1) | 92 (93.9) | 98 (41.5) | |
| Both | 0 (0) | 30 (100) | 30 (12.7) | |
| Hearing status | | | | |
| Well | 4 (3.7) | 104 (96.3) | | 0.758 |
| Decreased | 6 (4.7) | 122 (95.3) | 128 (54.2) | |
| Ear discharge type | | | | |
| White | 7 (5) | 133 (95) | 140 (59.3) | NA |
| Yellow | 0 (0) | 57 (100) | 57 (24.2) | |
| Bloody | 1 (4.3) | 22 (95.6) | 23 (9.7) | |
| Green | 2(12.5) | 14 (87.5) | 16 (6.8) | |
| Bottle feeding history | | | | |
| No | 5 (2.79) | 174 (97.21) | 179 (75.8) | |
| Yes | 5 (8.7) | 52 (91.2) | 57 (24.2) | 0.171 |
| Chronic disease | | | | |
| No | 4(1.79) | 219 (98.2) | 223 (94.5) | |
| Yes | 6(46.2) | 7 (53.8) | 13 (5.5) | 0.003 |
| Antibiotics use without Physician prescription | | | | |
| No | 4 (1.86) | 210 (98.1) | 214 (90.7) | |
| Yes | 6 (27.2) | 16 (72.7) | 22 (9.3) | <0.001 |
| Upper respiratory tract infection | | | | |
| No | 8(6.1) | 124(93.9) | 132(55.9) | |
| Yes | 2(1.9) | 102(98.1) | 104(44.1) | 0.199 |
| Previous hospital visit and treatment | | | | |
| No | 2(1.35) | 146(98.6) | 148(62.7) | |
| Yes | 8(9.1) | 80(90.9) | 88(37.3) | 0.006 |
| **Total** | **10 (4.2)** | **226 (95.8)** | **236 (100)** | |

NA: Not applicable, AOM: Acute otitis media, CSOM: Chronic suppurative otitis media.

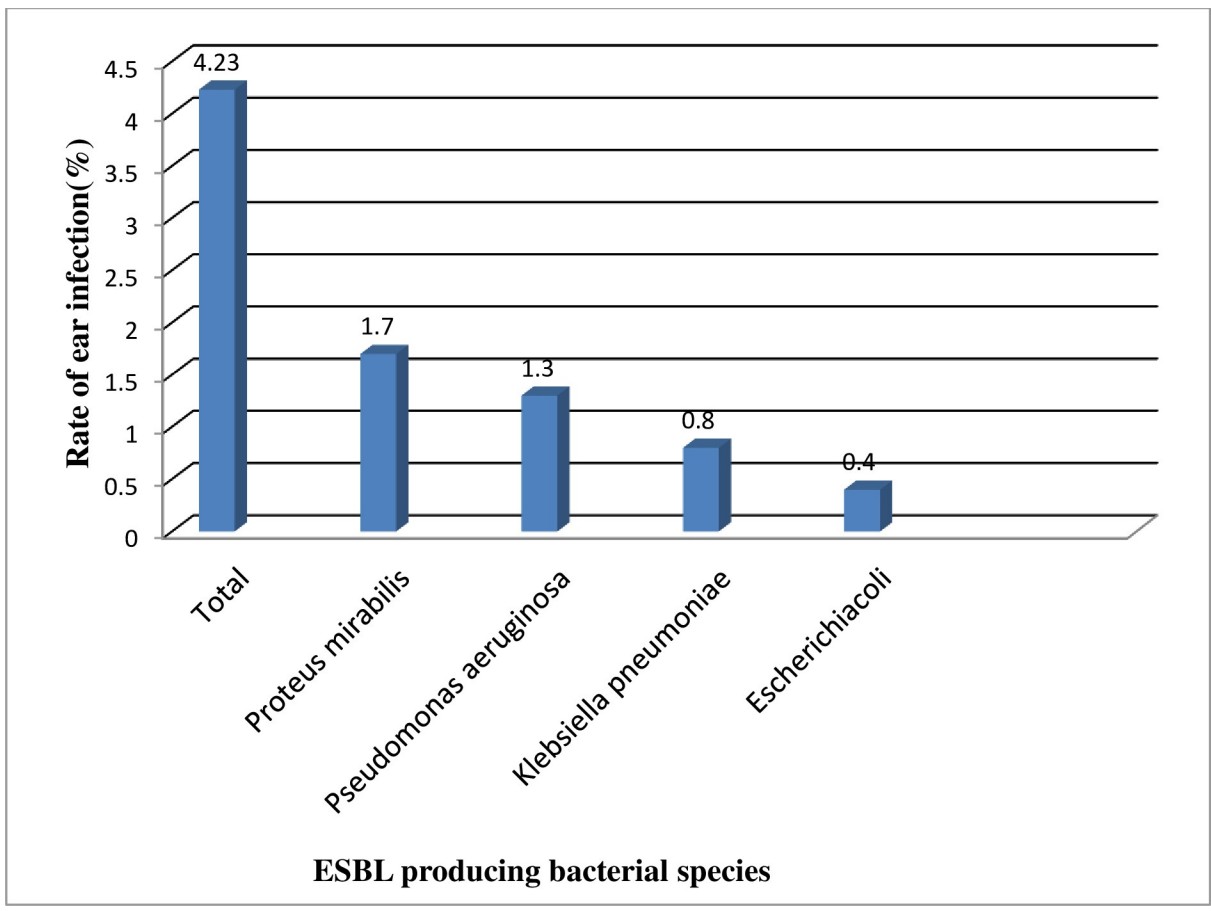

**Fig 1. Proportion of ear infection with individual ESBL producing bacterial isolates at FHRH, January, 2019 (N = 236).**

### Multi-drug resistance profiles of bacterial isolates

All ESBL producing bacteria isolated from ear infection were MDR (100%). However, the frequency of MDR among ESBL negative isolates was 48(35.5%) (P<0.001) (Table 4). Overall, 58 (43%) of the total isolates were MDR. The MDR profile of *P.aeruginosa*, *K.pneumoniae* and *P. mirabilis* were 29(53.7%), 8 (50%) and 15(34.9%), respectively (Table 5).

## Discussion

The prevalence of ear infection with ESBL producing bacteria was lower compared to other studies in Africa (8.3 and 16.3%) [17, 18], Meerut city (8.9%) [28] and India (18.3 and 74.2%) [16, 29]. However,astudy in Nigeria reported the absenceof ESBL producing bacteria inear infection [30]. This might be due to variations in the study participants, extent of antiobiotic use and ESBLs detection methods. The relatively lower proportion of ear infection with ESBL producing bacteria in the current studymight be linked with low risk of exposure to ESBL producing bacteria from hospital settingssince the participants were outpatients and most of themhad chronic cases came to FHRH hospital lately after the infection failed to heal by itself or local health care treament.

In the present study, ear infection with ESBL producing bacteria was significantly higher among age groups of 31–40 years (P = 0.032) compared with other age groups. This was consistent with the study conducted in India [29]. This might be due to increased number of

**Table 2. Proportion of ESBL producing bacteria from the total Gram negative isolates at FHRH, May 2018 to January 2019, (N = 135).**

| Bacterial species | ESBL production | | |
|---|---|---|---|
| | Positive N (%) | Negative N (%) | Total N (%) |
| *Klebsiella pneumoniae* | 2(14.3) | 12 (85.7) | 14 (10.4) |
| *Escherichia coli* | 1(12.5) | 7 (87.5) | 8 (5.9) |
| *Proteus mirabilis* | 4(10.8) | 33 (89.2) | 37 (27.4) |
| *Pseudomonas aeruginosa* | 3 (6.4) | 44 (93.6) | 47 (34.8) |
| *Enterobacter* species | 0 (0) | 12 (100) | 12 (8.9) |
| *Citrobacter* species | 0 (0) | 1(100) | 1(0.74) |
| *P.aeruginosa+ P.mirabilis* | 0 | 5 | 5 (3.7) |
| *P.aeruginosa +E.coli* | 0 | 1 | 1(0.74) |
| *P.mirabilis+K.pneumoniae* | 0 | 1 | 1(0.74) |
| *P.aeruginosa +K.pneumoniae* | 0 | 1 | 1 (0.74) |
| Total *P.aeruginosa* | NA | NA | 54 (40) |
| Total *P.mirabilis* | NA | NA | 43(31.9) |
| Total *K.pneumoniae* | NA | NA | 16(11.9) |
| Total *E.coli* | NA | NA | 9 (6.7) |
| **Total isolates** | **10 (7.4)** | **117 (86.7)** | **135 (57.2)** |

NA: Not applicable.

CSOM in this age group related with delayed and inadequate treatment. Poor housing and living conditions and poor access to medical care contribute to chronicity of the illness which might increase the acquisition of ESBL [31, 32].

In the current study all ear infections with ESBL producing bacteria were from rural residents. This is because patients from rural residents use non-sterile inanimate objects to remove ear wax.This holds true that ESBL producing bacteria exists on any environmental surfaces. Moreover, it might be also related with late and emperical treatment of cases [31, 32]. At the same time, the proportion of ear infection with ESBL producing bacteria was high in patients with other chronic illness. The result concurs with a report from Saudi Arabia [33].The role of chronic disease in the presence of ESBL enzyme might be linked with patients' frequent

**Table 3. Antimicrobial resistance profiles of Gram negative bacteria isolated from ear discharge swabs at FHRH, May 2018 to January 2019 (N = 135).**

| Antimicrobials | *P.aeruginosa* | | *P.mirabilis* | | *K. pneumoniae* | | *E.coli* | | *Enterobacter* spp. | | *Citrobacter* spp. | | Total | |
|---|---|---|---|---|---|---|---|---|---|---|---|---|---|---|
| | #T | R% | #T | R% | #T | R% | #T | R% | #T | R% | #T | R% | #T | R% |
| Cotrimoxazole | 54 | 17 (31.5) | 43 | 18 (41.9) | 16 | 5 (31.2) | 9 | 0 (0) | 12 | 1 (8.3) | 1 | 1 (100) | 135 | 42(31.1) |
| Ciprofloxacin | 54 | 3 (5.6) | 43 | 3 (6.9) | 16 | 1 (6.3 | 9 | 0 (0) | 12 | 0 (0) | 1 | 0 (0) | 135 | 7 (5.2) |
| Ampicillin | 54 | 52 (96.3) | 43 | 40 (93.0) | 16 | 14 (87.5) | 9 | 8 (88.9) | 12 | 11 (91.7) | 1 | 1 (100) | 135 | 126 (93.3) |
| Amoxicillin-Clavulanic acid | 54 | 40 (74.1) | 43 | 20 (46.5) | 16 | 9 (56.3) | 9 | 4 (44.4) | 12 | 6 (50.0) | 1 | 0 (00 | 135 | 79 (58.5) |
| Ceftriaxone | 54 | 12 (22.2) | 43 | 5 (11.6) | 16 | 4 (25.0) | 9 | 3 (33.3) | 12 | 3 (25.0) | 1 | 0 (0) | 135 | 27 (20) |
| Ceftazidime | 54 | 9 (16.7 | 43 | 5 (11.6) | 16 | 4 (25.0) | 9 | 1 (11.1) | 12 | 1 (8.3) | 1 | 0 (0) | 135 | 20 (14.8) |
| Aztreonam | 54 | 11 (20.4 | 43 | 5 (11.6) | 16 | 4 (25.0 | 9 | 1 (11.1) | 12 | 1 (8.3) | 1 | 0 (0) | 135 | 22 (16.3) |
| Gentamicin | 54 | 16 (29.6) | 43 | 7 (16.3) | 16 | 5 (31.2) | 9 | 0 (0) | 12 | 1 (8.3) | 1 | 0 (0) | 135 | 29 (21.5) |
| Cefotaxime | 54 | 6 (11.1) | 43 | 4 (9.3) | 16 | 4 (25.0) | 9 | 1 (11.1) | 12 | 1 (8.3) | 1 | 0 (0) | 135 | 16(11.9) |
| **Total** | **486** | **166 (34.2)** | **387** | **107 (27.6)** | **144** | **50 (34.7)** | **81** | **18 (22.2)** | **108** | **25 (23.1)** | **9** | **2 (22.2)** | **1215** | **368 (30.3)** |

#T: Number of bacterial isolates tested, R%: number and percent of bacterial isolates resistance to the antibiotic.

**Table 4. MDR profiles among ESBL producing and non-ESBL producing isolates from ear discharge swabs at FHRH, May 2018 to January 2019.**

| Bacterial isolate | MDR, N (%) | | | | | | | | |
|---|---|---|---|---|---|---|---|---|---|
| | R0 | R1 | R2 | R3 | R4 | R5 | ≥R6 | MDR | P-value |
| **ESBL positive** | 0 (0) | 0 (0) | 0 (0) | 0 (0) | 1 (10) | 2(20) | 7(70) | 10 (100) | P<0.001 |
| **ESBL Negative** | 6(4.4) | 31(22.9) | 40(29.6) | 26(19.3) | 13 (9.6) | 2 (1.5) | 7 (5.2) | 48(35.5) | |

R0: Suceptible to all antibiotics; R1-7: resistance to 1, 2, 3, 4, 5, 6, and 7 antibiotics; MDR: Multi Drug Resistance of the isolate against ≥ 3 antibiotics from different classes.

exposure to antibiotics for the management of opportunistic infections which causes for selection and emergence of drug resistance.

The frequency of ear infection with ESBL producing bacteria was significantly high among patients who took antibiotics without physician prescription in the current study (P<0.001). This is coherent with the study done in Thailand [34]. Misuse of antibiotics is one of the major reasons for selection of resistant mutants. Moreover, patients with ear infection are usually treated without culture isolation and antibiotic susceptibility testing in the study area. Thus, the chance of treating cases of ear infection with unnecessary antibiotics is high.

In the present study, *P.mirabilis* was the major cause of ear infection with ESBL production. This was in agreement with similar studies conducted in Nigeria [17] and India [29]. Moreover, ESBL producing *P.aeruginosa* was the second leading cause of ear infection. However, its proportion (1.3%) was lower than reports from India (23.2%) [29] and higher than from Nigeria [30].The predominance of ESBL producing *P.mirabilis* and *P.aeruginosa* isolates compared to others might be due to high proportion of these bacteria in ear infection. Moreover, *P.mirabilis* and *P.aeruginosa* are widely distributed in the places where poor sanitary conditions exist. This might contributes the transfer of ESBL resistance genes in the isolates [33, 34].

The overall proportion of ESBL producing species in the present study (7. 4%) was lower than reports from Nigeria (40.2%) [18], Tanzania (42%) [19] and India (69.1%) [29]. This might be due to variation in patients type and guidelines of ESBL detection methods. Phenotypic ESBL detection methods have different guidelines and the use of variable guidelines could result on variability in the magnitude of ESBL production for different studies. Moreover, bacterial species that are susceptible for acquiring ESBL genes are different in different parts of the world.

**Table 5. Antibiogram of bacteria isolated from ear discharge swab at FelegeHiwot Referral Hospital from May 2018 to January 2019, North West Ethiopia (N = 135).**

| Isolated bacteria | Antibiogram, N (%) | | | | | | | |
|---|---|---|---|---|---|---|---|---|
| | R0 | R1 | R2 | R3 | R4 | R5 | ≥ R6 | MDR |
| *P. aeruginosa* (n = 54) | 2 (3.7) | 6 (11.1) | 17 (31.5) | 13 (24.1) | 8 (14.8) | 2 (3.7) | 6 (11.1) | 29 (53.7) |
| *P. mirabilis* (n = 43) | 2 (4.7) | 12 (27.9) | 14 (32.6) | 6 (13.9) | 5 (11.6) | 2 (4.7) | 2 (4.6) | 15 (34.9) |
| *K. pneumoniae* (n = 16) | 1 (6.3) | 4 (25) | 3 (18.8) | 4 (25) | 2 (12.5) | 1 (6.3) | 1 (6.3) | 8 (50) |
| *Enterobacter*species (n = 12) | 0 (0) | 6 (50) | 2 (16.7) | 3 (25) | 0 (0) | 0 (0) | 1 (8.3) | 4 (33.3) |
| *E. coli* (n = 9) | 1 (11.1) | 3 (33.3) | 3 (33.3) | 1 (11.1) | 0 (0) | 0 (0) | 1 (11.1) | 2 (22.2) |
| *Citrobacter* species (n = 1) | 0 (0) | 0 (0) | 1 (100) | 0 (0) | 0 (0) | 0 (0) | 0 (0) | 0 (0) |
| Total (135) | 6 (4.4) | 31 (23) | 40 (29.6) | 27 (20) | 15 (11.1) | 5 (3.7) | 11 (8.1) | 58 (43) |

R0: resistance to no antibiotics; R1-7: resistance to 1, 2, 3, 4, 5, 6, and 7 antibiotics; MDR: Multi Drug Resistance of the isolate against ≥ 3 antibiotics from different classes.

In this study, the highest proportion of ESBL production (14.5%) was recorded for *K.pneumoniae* species. The proportion of *K.pneumoniae* is slightly higher than similar studies done in Nigeria (4.9%) [17] and India (3%) [29].

ESBL producing *K.pneumoniae* is one of the currently priority pathogens for intervention and response. *K. pneumoniae* strains are easily spreading, efficient at acquiring and disseminating resistance plasmids and production of different types of ESBLs [35].

In the current study, *E.coli*, *P.aeruginosa* and *P.mirabilis* were ESBL producing with a proportion of 12.5%, 6.8% and 10.8%, respectively. This was consistent with a study in Meerut City where isolates of *P.mirabilis* and *E.coli* were ESBL producing, respectively [28].However, 8 out of 16 isolates of *P.mirabilis* was ESBL producers according to a report from Iraq [36].

In this study, *P. aeruginosa* showed high level of resistance to amoxicillin-clavulanic acid (74.1%) and ampicillin (96.3%).This was consistent with previous reports from Bahir Dar (89 and 90.9%) [3], Mekelle (88.9 and 100%) [12] and Tanzania (95 and 100%,) [19], respectively. The high level of *P.aeruginosa* resistance to these antibiotics might be due to biofilm formation. Biofilms which are firmly adhered to the damaged tissue of middle ear mucosa are impermeable to antibiotics [35]. Besides, its intrinsic and acquired resistance via outer membrane impermeability, the presence of numerous genes coding resistance to different classes of antibiotics contributes this high resistance rate in *P. aeruginosa* [37]. Moreover, the high rate of *P. mirabilis* resistance to ampicillin (93%) in the current study is also consistent with reports from Bahir Dar (87.8%) [3], Mekelle (64%) [12] and Tanzania (100%) [19]. This might be due to the frequent empirical treatment of patients which results the development of antibiotic resistance in Enterobacteriaceae [4].

In the present study all bacterial isolates showed low level of resistance to ciprofloxacin (0–6.9%), and gentamicin (0–29.6%).This result is in agreement with the study by other authors in Africa where 0–33% and 0–37.5% level of resistance were reported against ciprofloxacin and gentamicin, respectively [4, 19].Gram negative species of the present study also revealed low level of resistance against ceftazidime, ceftriaxone and cefotaxime. This was comparable with previous report in Ethiopia [38]. Moreover, only 16% level of resistance was reported against aztreonam in the current study. This was lower than a report from other part of Ethiopia [38].The low rate of resistance against 3<sup>rd</sup> generation cephalosporins and monobactams might be due to these drugs are not commonly prescribed for the management of ear infection in the study area. Therefore, these antibiotics should be wisely used to prevent the development of resistance in Gram negative bacilli.

The proportion of MDR among ESBL producing species was 100% in the present study. This was consistent with reports from Libya [34] and Korea [37]. ESBL producing bacteria carry genes encoding for resistance to other classes of antibiotics such as aminoglycosides, quinolones and sulfamethoxazole-trimethoprim. Thus, ESBL-producing organisms often possess an MDR phenotype than non ESBL producers [14]. This finding collectively showed the rising of ESBL producing MDR isolates in developing countries which might be attributed to a loose control of antibiotic utilization and lack of appropriate diagnosis of antimicrobial resistance for accurate treatment.

The overall MDR rate of Gram negative isolates in this study (43%) was lower than studies done in Mekele (74.5%) [12]and Jimma (67%) [4],Ethiopia. This might be due to variations in the classes of antibiotics, MDR definition, specimen type and study participants. Repeated, inappropriate and incorrect use of antimicrobial agents in empirical treatment might be some of the different factors that can contribute to the development of MDR among Gram negative bacilli solates from ear infection. In this study, MDR was high in *P.aeruginosa* isolates (53.7%). This result was in tandem with reports from Egypt (66.6%) [39].This high MDR rate of *P.aeruginosa* might due to its intrinsic resistance to many classes of antimicrobial agents and ability

to acquire resistance by mutation and horizontal transfer of resistance determinants. Resistance of *P.aeruginosa* is also usually accompanied by the production of beta lactamases, active expulsion of antibiotics by efflux pump, and alteration of outer-membrane protein expression [39].

### Limitations of the study

In this study use of double disk synergy test to confirm ESBL producer might not exclude other beta-lactamases and co-existence of these different beta lactamase enzymes might mask detection of ESBL. Moreover, some bacteria might produce inhibitor resistance variant of ESBL which is difficult to differentiate by the method used in this study.

### Conclusions

Ear infection due to ESBL producing bacteria coupled with high levels of MDR becomes a major concern for outpatients. *P.mirabilis* was the leading causes of ear infection with ESBL production followed by *P.aeruginosa*. Most of Gram negative isolates showed high level of resistance to commonly prescribed antibiotics such as ampicillin and amoxicillin-clavulanic acid. However, ciprofloxacin, third generation cephalosporins and aztreonam were effective for most bacterial isolates. Therefore, treatment of ear infection in the study area needs to be guided by laboratory confirmation. Moreover, regular antibiotic resistance surveillance is required to detect and respond the spreading of ESBL and carbapenemase producing bacterial pathogens.

### Supporting information

**S1 File.**
(DOCX)

### Acknowledgments

Our gratitude also goes to staff members of FHRH and its Ear, Nose and Throat clinic for their permission to conduct this research and contribution during data and specimen collection. We also acknowledge Amhara Public Health Institute for allowing us to carry out the laboratory work in their setting. We would like to thank all APHI microbiology staffs for their unreserved technical support. We are also grateful to patients from which the clinical data and samples were collected.

### Author Contributions

**Conceptualization:** Kindye Endaylalu, Bayeh Abera, Wondemagegn Mulu.

**Data curation:** Bayeh Abera, Wondemagegn Mulu.

**Formal analysis:** Kindye Endaylalu, Wondemagegn Mulu.

**Investigation:** Kindye Endaylalu, Wondemagegn Mulu.

**Methodology:** Kindye Endaylalu, Bayeh Abera, Wondemagegn Mulu.

**Project administration:** Bayeh Abera, Wondemagegn Mulu.

**Resources:** Kindye Endaylalu, Wondemagegn Mulu.

**Software:** Kindye Endaylalu, Wondemagegn Mulu.

**Supervision:** Bayeh Abera, Wondemagegn Mulu.

**Validation:** Kindye Endaylalu, Bayeh Abera, Wondemagegn Mulu.

**Visualization:** Bayeh Abera, Wondemagegn Mulu.

**Writing – original draft:** Wondemagegn Mulu.

**Writing – review & editing:** Kindye Endaylalu, Bayeh Abera, Wondemagegn Mulu.

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
