## [Decision Letter · Decision Letter 0]

6 Aug 2020

PONE-D-20-18371

Extended spectrum beta lactamase producing bacteria profiles among outpatients with ear infection at Felege Hiwot Referral Hospital, North West Ethiopia

PLOS ONE

Dear Dr. Mulu,

Thank you for submitting your manuscript to PLOS ONE. After careful consideration, we feel that it has merit but does not fully meet PLOS ONE’s publication criteria as it currently stands. Therefore, we invite you to submit a revised version of the manuscript that addresses the points raised during the review process.

Regarding guideline used for antimicrobial susceptibility, please indicate the guideline used- CLSI or EUCAST

We look forward to receiving your revised manuscript.

Kind regards,

Iddya Karunasagar

Academic Editor

PLOS ONE

Additional Editor Comments:

Please revise the manuscript addressing all points raised by the reviewers.

Journal Requirements:

3. Please provide a copy of the questionnaire used in your study as a supporting information file.

5. We note you have included a table to which you do not refer in the text of your manuscript. Please ensure that you refer to Table 4 in your text; if accepted, production will need this reference to link the reader to the Table.

Reviewers' comments:

Reviewer's Responses to Questions

**Comments to the Author**

1. Is the manuscript technically sound, and do the data support the conclusions?

Reviewer #1: Yes

2. Has the statistical analysis been performed appropriately and rigorously? 

Reviewer #1: Yes

3. Have the authors made all data underlying the findings in their manuscript fully available?

Reviewer #1: Yes

4. Is the manuscript presented in an intelligible fashion and written in standard English?

Reviewer #1: Yes

5. Review Comments to the Author

Reviewer #1: 1. Introduction can be made more concise

2. Line 3- sentence looks incomplete

3. Line 41-43- probably authors mean factors associated with ear infections

4. Line105-106-please check the sentence

5. Line113- 114- one of the reference either CLSi or Eucast can be used instead of both

6. Line 117-118- different generation cephalosporin could have been put for susceptibility.

7. Line 123- 130- Please put reference

8. Line 167- please reframe sentence for better understanding

9.Line 287- please check the sentence in in view of the fact that esbl producers are resistant to monobactams

6. PLOS authors have the option to publish the peer review history of their article (what does this mean?). If published, this will include your full peer review and any attached files.

Reviewer #1: No

---

## [Author Response · Author response to Decision Letter 0]

21 Aug 2020

Point by point responses to reviewers comment

PONE-D-20-18371

Title of the manuscript: Extended spectrum beta lactamase producing bacteria profiles among outpatients with ear infection at Felege Hiwot Referral Hospital, North West Ethiopia

PLOS ONE

Dear Editor,

We have found the editors’ remarks and comments of the reviewers valuable. We would like to thank you for considering our manuscript for publication to your journal. We have revised the manuscript based on your comment. Regarding the remarks from the editor, we have thoroughly revised the manuscript to improve its readability and all the other comments given by reviewers are incorporated in the revised version of the manuscript. The manuscript is thoroughly copyedited for language usage, spelling and grammar by our colleague. All grammatical and typographical errors are corrected and the changes made are highlighted. The manuscript was also checked and revised for compliance with PLOS ONE's style requirements. 

Point –by –point responses to Editors comment

Regarding guideline used for antimicrobial susceptibility, please indicate the guideline used- CLSI or EUCAST

Response: Well taken and only CLSI guideline was used. This is corrected in the revised manuscript.

Additional Editor Comments:

Please revise the manuscript addressing all points raised by the reviewers.

Response: Well taken and all points raised by reviewers are addressed in the revised manuscript

 Journal Requirements:

 Response: Well taken. We have strictly followed the pLOS ONE style requirements. 

Response: Well taken and the manuscript is thoroughly copy edited for the language usage, spelling and grammars by the colleague Professor Mulugeta Kibret. 

3. Please provide a copy of the questionnaire used in your study as a supporting information file.

Response: The soft copy of the questionnaire used in both English and local language is uploaded as a supporting information file.

 Response: Well taken and corrected in the revised manuscript

5. We note you have included a table to which you do not refer in the text of your manuscript. Please ensure that you refer to Table 4 in your text; if accepted, production will need this reference to link the reader to the Table.

Response: Well taken and Table 4 is referred in the text of the revised manuscript

Reviewers' comments:

Point by point responses to reviewers comment

Reviewer #1: 

1. Introduction can be made more concise

Response: We have revised the introduction but the size is still not changed.

2. Line 3- sentence looks incomplete

Response: Well taken and the incomplete sentence completed in the revised manuscript

3. Line 41- 43- probably authors mean factors associated with ear infections

Response: Well taken and the phrase ESBL production was substituted by ear infection in the revised manuscript.

4. Line105-106-please check the sentence

Response: Checked and minor revisions made. 

5. Line113- 114- one of the references either CLSi or Eucast can be used instead of both

Response: Well taken. Only CLSI guideline referred in the revised manuscript. This is corrected both in the manuscript and list of references.

6. Line 117 - 118- different generation cephalosporin could have been put for susceptibility.

Response: Only third generation cephalosporins were used in the investigation and this specified in the revised manuscript

7. Line 123- 130- Please put reference

Response: Well taken and a reference number 27 is cited in the revised manuscript

8. Line 167- please reframe sentence for better understanding

Response: Well taken and corrected in the revised manuscript

9.Line 287- please check the sentence in in view of the fact that esbl producers are resistant to monobactams.

Response: Well taken and checked in the revised manuscript

---

## [Editor Report · Decision Letter 1]

26 Aug 2020

Extended spectrum beta lactamase producing bacteria among outpatients with ear infection at FelegeHiwot Referral Hospital, North West Ethiopia

PONE-D-20-18371R1

Dear Dr. Mulu,

We’re pleased to inform you that your manuscript has been judged scientifically suitable for publication and will be formally accepted for publication once it meets all outstanding technical requirements.

Kind regards,

Iddya Karunasagar

Academic Editor

PLOS ONE

Additional Editor Comments (optional):

The authors have addressed all comments.
---

## [Editor Report · Acceptance letter]

31 Aug 2020

PONE-D-20-18371R1 

Extended spectrum beta lactamase producing bacteria among outpatients with ear infection at FelegeHiwot Referral Hospital, North West Ethiopia 

Dear Dr. Mulu:

I'm pleased to inform you that your manuscript has been deemed suitable for publication in PLOS ONE. Congratulations! Your manuscript is now with our production department. 

Kind regards, 

on behalf of

Dr. Iddya Karunasagar 

Academic Editor

PLOS ONE